# Promoting Enquiry Skills in Trainee Teachers within the Context of the University Ecological Garden

**Lourdes Aragón ***[ID] and Beatriz Gómez-Chacón [ID]

Department of Didactics, Area of Didactics of Experimental Sciences, Faculty of Education Sciences, University of Cádiz, 11519 Cádiz, Spain; beatriz.gomezchacon@uca.es
* Correspondence: lourdes.aragon@uca.es

**Abstract:** One of the objectives of science teaching and learning is to achieve quality science education, which involves improving initial teacher training. The use of methodologies that promote learning in science, such as the enquiry-based learning strategy, are encouraged. It is also necessary to provide appropriate contexts that give meaning to the investigation conducted, and arouse the students' interest. The purpose of this study is to identify the skills related to the enquiry competency that future pre-school teachers acquire after carrying out investigations using the University Ecological Garden as a context. To undertake this study, a non-experimental quantitative methodology was developed based on the application of two instruments: the New Practical Test Assessment Inventory (NPTAI), based on the Practical Test Assessment Inventory, and the trainee teachers' Enquiry Competency Level (ECL), adapted for the present work. Thirty-seven group reports were analysed and recoded to establish five levels of enquiry competency. A predominance of students with a high level of enquiry competency as opposed to "pre-scientific" and "unscientific" lower levels was observed. The results allowed us to explore the role of the teacher in the monitoring process during the strategy, the context used, and the main difficulties encountered in the implementation of the strategy.

**Keywords:** ecological garden; enquiry; initial teacher training; competency level

## 1. Introduction

To improve science teaching and learning, and therefore, to achieve quality scientific training, an in-depth renewal of initial teacher training is required [1]. According to Vilches and Gil [2], one of the most appropriate strategies would be to encourage future teachers to learn contents that are specific to their subject through a process of investigation and immersion in scientific culture. The enquiry-based strategy (EBS) is considered a relevant methodology to achieve this change [1,3].

To implement the EBS in the classroom, teachers need to have adequate knowledge, skills and habits of thought. Otherwise, it is unlikely they will be motivated or proactive enough to undertake genuine enquiry processes and engage their students in said processes [4]. According to Toma et al. [5], future teachers implement this teaching approach insofar as they are able to formulate scientifically oriented questions. Said questions allow students to offer explanations based on evidence that can subsequently be communicated in a justified manner. In the case of the early childhood education stage, there is a broad consensus on the idea that scientific training should start from an early age [6,7]. However, in Spain, there is little scientific evidence of educational practices aimed at achieving a quality EBS [8]. In this regard, Cantó et al. [9] carried out a study based on the perceptions of future teachers during their internship period. Serious shortcomings were identified in how science is taught in early childhood education classrooms, and it was concluded that few activities characteristic of scientific processes were being undertaken. These results show the need to train future teachers in these teaching methodologies, taking into account their relevance for children to develop cognitive-linguistic skills that are specific to science,

such as identifying, describing, defining, explaining and justifying from an early age. Their enormous importance in the construction of the so-called precursory scientific mental models [10–12] is also worth noting. According to Lemeignan and Weil-Barrais [13], these precursory models are key cognitive structures generated in school contexts, and they are the foundation for building other more advanced mental models throughout successive educational stages.

This dissociation between theory and practice in initial teacher training is not only a relevant, but also a worrying issue [14]. Cantó et al. [9] point out that in initial teacher training, educators cannot pass on contradictory and incoherent discourses regarding the educational model that trainee teachers should develop in their future teaching, and the one they later observe in the classroom when completing their internship period. According to Korthagen and Kessels [15], educators of future teachers should provide their students with genuine learning experiences that promote greater reflection on and awareness of the teaching models they are developing.

Another crucial issue to consider is the training of teachers to undertake enquiry processes in the classroom. Planning an EBS requires teachers to have appropriate professional skills for its implementation in the classroom to be successful. They need to have sufficient theoretical knowledge of the characteristics of the EBS, as well as pedagogical and methodological skills to use it in the classroom. A positive attitude towards applying the EBS in their teaching practice [16] is also necessary. According to Shanmugavelu et al. [17], in order to draw conclusions, this research strategy also requires higher-order thinking skills and critical thinking skills.

Teacher trainers specifically need to improve those training needs that correspond to the "knowledge bases" of teaching, proposed by Shulman in the 1980s [18]. According to Nitz et al. (2010, cited in Gairín et al. [19]), after more than twenty years of research, there is a broad consensus regarding professional teaching knowledge. It is organised into three categories: (1) content knowledge (CK), which [20] considers both the mastery of those principles, theories, structures and theoretical frameworks of the discipline to be taught, and knowing how this knowledge is generated, its epistemology and how it is communicated; (2) pedagogical content knowledge (PCK), which refers to the body of knowledge related to how science is learnt, the curriculum and different types of teaching and assessment strategies in order to transform scientific knowledge into effective teaching; and (3) general pedagogical knowledge (GPK), which refers to those broad principles and strategies of classroom management and organisation that appear to transcend subject matter [21]. From the Didactics of Sciences, Solbes et al. [22] (p. 27) establish a possible model of professional teaching knowledge in accordance with the previous approaches and which is specified in the following points:

1. Content knowledge in a broad sense, which also includes knowing the history of science, the nature of science (NOS), and the methodologies scientists use in their work; science, technology and society (STS) interactions, and the ability to select and sequence the appropriate didactic content.
2. Pedagogical knowledge, including classroom management, the use of ICT, etc.
3. Pedagogical content knowledge, considered as orientations and conceptions about the teaching of science, teaching strategies (including the EBS), knowledge of learning and ideas of the students, assessment methods, knowledge of the curricula and learning materials, etc.

Applying an EBS in real contexts combines the possibility of formulating questions that generate curiosity and motivation in students, and helps make sense of the investigation proposed [23]. Real contexts, such as an ecological vegetable garden, are considered suitable spaces, since they constitute true living laboratories in which to observe and experiment with their components and processes. This promotes the connection between theory and practice of contents linked to science [24]. The context of a vegetable garden is also considered a space for social development related to health [25].

Within this framework, the main objective of this study is to identify the skills related to the enquiry competency level (ECL) acquired by future pre-school teachers after designing and performing small contextualised investigations in the University Ecological Garden (UEG). These investigations take place within the compulsory subject of Didactics of Experimental Sciences.

## 2. Theoretical Framework

### 2.1. The EBS in Science Teaching and Learning

The EBS is defined as a learning strategy that brings students closer to how scientists study nature by proposing explanations based on evidence. This strategy evolved from proposals made in the 1920s and 1970s by authors such as Dewey [26] and Schwab [27]. Said proposals arose in response to science education focused on accumulating information, and not on developing attitudes and skills. They emerged because the very nature of science as knowledge is not static, and it often needs to be revised and updated as a result of new findings. The term enquiry has two different meanings. According to Abd El-Khalick et al. [28], enquiry refers to those skills students should develop to be able to carry out scientific research and work as scientists do to solve problems. Enquiry may also refer to those teaching and learning strategies that allow science to be learnt from conducting investigations that provide experimental evidence in order to promote the creation and development of school scientific knowledge. This study is concerned with its second meaning—in other words, with how to include enquiry in the classroom as a specific educational strategy for science teaching, or as a didactic model to learn science through enquiry [29].

From this perspective, the EBS is situated within a constructivist framework of learning and is based on the active learning of students. It allows students to acquire skills and competencies characteristic of science, and to develop autonomy and critical thinking skills with regard to scientific knowledge [30]. Barrow [31] states that, when carrying out an enquiry, we promote the development of teaching strategies that motivate learning, as well as questioning and experimental skills. Conducting an enquiry thus implies strengthening skills such as observation, asking questions, systematising data, formulating hypotheses, as well as reflection on the analysis and interpretation of data [28,32,33].

The social and cultural transformations of today's society require reflection, and, as a result, changes need to be proposed in the didactic methodology teachers should use in the classroom to meet those needs. Several studies reveal that many teachers continue to use transmission–reception models, while those who choose models based on enquiry are still a minority [34–36]. The methodologies employed in the classroom should provide students with tools and skills that increase their autonomy. Means and resources that facilitate their future work in the classroom are also necessary. As pointed out by López [37], the university teacher should lead a shift in education towards the search for new strategies that promote the development of creativity, quality, competencies and collaboration.

In faculties of education, this need is apparent because trainee teachers pass on their experiences to their future classrooms. New ways of understanding teaching and learning processes should therefore be acquired. To that end, it is essential for trainee teachers to previously acquire skills regarding the design and development of activities aimed at gaining knowledge to propose methods, strategies and techniques to achieve a more global and interdisciplinary objective. A balance between knowledge, skills and abilities should prevail in initial teacher training. It is necessary to combine strategies that promote scientific knowledge in the teaching process to address both personal and professional situations [38].

When using the EBS, students experience the work of scientists first-hand, in addition to becoming familiar, by conducting simple investigations, with scientific work and with the manner in which results are obtained. According to Harlen [39], the EBS carries a series of implications for the students, such as learning activities. These activities need to engage students, as well as allow them to develop and use scientific skills. The activities have to

include dialogue and discussion both with peers and teachers, collaborative work, and building new knowledge together.

If we focus on the level of student engagement in the investigative process, four types of enquiry are identified [40]: (1) Open enquiry, where the students design the entire investigative process through their own problem question for investigation, and make decisions; (2) Guided enquiry, in which the problem question for investigation is provided by the teacher guiding the investigation by means of a set of questions; (3) Coupled enquiry, an intermediate stage between the previous two, since the question to investigate is provided by the teacher, but the students plan the investigation following their own decisions; and (4) Structured enquiry, in which the teacher leads the investigation, giving indications of the steps to be followed.

To implement the EBS in the classroom, different phases simulating the scientific process are considered. Martínez-Chico [41] establishes the following:

1.  Identify problems or questions of a scientific nature (the answers to which can be confirmed or rejected using evidence). Open problematic situations that are of interest to students can be taken as a starting point.
2.  Construct hypotheses as possible answers to the problem or question (justified explanations). Formulating hypotheses takes into account the students' preconceptions and mental models.
3.  Look for evidence that confirms or rejects the hypothesis (through experiments or search of information).
4.  Analyse and interpret results. The analysis of results is performed in light of the body of knowledge available, and is appropriate to the level of the students.
5.  Draw conclusions and communicate them. Both the results and the conclusions can be communicated by means of reports, in which attention is paid to the development of oral and written communication skills. Throughout the entire process, the collective dimension of the work is enhanced.

The implementation of these phases in the classroom allows for developing certain didactic assumptions structured into organised sequences of learning activities [42].

### 2.2. The University Ecological Garden (UEG) as a Learning Context in Initial Teacher Training

The use of vegetable gardens as a didactic tool and context is part of an active strategy called "garden-based learning." According to Desmond et al. [43], this strategy includes a whole set of programmes, activities and projects carried out in the garden as a basis for integrated learning. It links several disciplines, and implies an active and motivating experience connected to the real world. Its use in school contexts dates back to the 1890s, led by educators such as John Dewey. He put forward a more approachable didactic orientation between the limits of learning in the classroom and the natural environment used, for instance, the school vegetable garden [44]. From then onwards, and until the beginning of the 20th century, school gardens started to proliferate.

Several studies on the UEG as a learning context in initial teacher training have been carried out in Spain in recent years. The UEG is approached from very different perspectives [45–50]. In some cases, the garden is used as a multi-disciplinary resource in the development of curricular practices in different subjects of the degrees in Early Childhood and Primary Education [51]. Other educational experiences, such as those carried out by Eugenio and Aragón [24] at Universidad de Valladolid and Universidad de Cádiz, are aimed at promoting scientific training or working on learning related to environmental education.

Despite being a well-established resource in universities, it is necessary to guide investigations around the development and acquisition of learning from the UEG in the context of teacher training. Trainee teachers perceive it as a valuable tool from a didactic point of view [52]. Many of them express the value of the vegetable garden as a significant didactic resource, and reflect on its possible benefits and contributions to their future professional practice [53]. However, several research studies show certain contradictions

when future teachers design didactic proposals related to the garden, which confirms the difficulty of integrating learning into their didactic models. Aragón [54] encountered some weaknesses in the didactic designs proposed by 49 students of the degree in Early Childhood Education (DECE) with respect to this resource. This study shows that, despite "experiencing" the EBS, a large number of students used the garden in the final phase of their didactic sequences, including activities to be carried out in this space, but they were not linked to the development of scientific skills. Neither did they use the vegetable garden as the cornerstone of their proposals.

In this paper, we start from the premise that the UEG is a learning space with a high didactic potential that allows for developing all scientific dimensions, such as conceptual and procedural knowledge, as well as a positive attitude towards science, in an integrated manner [55]. However, according to Williams and Dixon [56], the significant increase in the use of vegetable gardens from a wide diversity of approaches and perspectives has not been accompanied by rigorous and systematic research that evidences and helps understand the results of student learning through this resource at different educational stages.

## 3. Materials and Methods

### 3.1. Context and Participants

This study is part of a didactic proposal designed in the subject of Didactics of the Natural Environment (DNE) in the 3rd year of the DECE at Universidad de Cádiz, Spain, during academic year 2019–2020. It is a compulsory subject worth 8 ECTS credits taught in the first semester [57].

The DNE subject is structured into three large content blocks [58]. This study is part of block 3, which is aimed at the teaching and learning processes of science. In this block, the students, organised in working groups of three to six members, design and develop small investigations within the context of the UEG following a guided EBS. For this study, the teachers proposed problem questions to start the investigations, which are shown in Table 1 [59]. The problem questions were randomly assigned to the different working groups, but the students were allowed to exchange them with other groups. Each problem question was related to one or several general topics established by the early childhood curriculum in the region of Andalusia [60].

Block 3 lasted four weeks, for a total of eight sessions of one and a half hours each. Two work spaces were combined in the sessions: the classroom and the vegetable garden of the faculty of Education Sciences, located in a small inner patio (Figure 1).

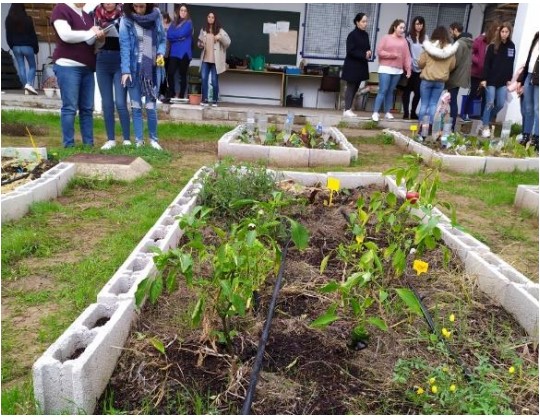

**Figure 1.** UEG of the faculty of Education Sciences at Universidad de Cádiz (Spain).

A total of 191 students, 10 male and 181 female students, aged 19 to 49, from three different class-groups (A, B and C) of the third year of the DECE participated in this research. The participants were organised into 37 working groups, which consisted of 4 to 6 members each.

**Table 1.** Distribution of the problem questions provided by the teachers to the working groups during the EBS.

| Early Childhood Education Curriculum Topics | Problem Questions in the UEG Context | Groups Assigned |
|---|---|---|
| living beings | Which is the most effective: earthworm compost or chemical fertiliser? A study using plants | GA9, GB9, GC9 |
| matter | How are material cycles closed in the vegetable garden? The three Rs in the ecological garden: decomposition columns | GA7, GC10 |
| water | Which irrigation systems are more efficient? A comparative study of different irrigation systems | GA4, GB12, GC4 |
| meteorological phenomena | How does air pollution affect plant growth? | GA3, GB1, GC5 |
| matter, water | How does the type of soil contribute to water retention? | GA10, GB7, GC1 |
| matter | How can we improve soil fertility in our vegetable garden? | GA6, GA12, GB8, GC3, GC8 |
| living beings, vital functions | How can acid pollution affect photosynthesis in plants? | GA11,GB4 |
| living beings, matter | How does the use of plant cover contribute to soil protection? | GA2,GB5, GC12 |
| different environments, matter, living beings | Which soil is best for the garden? Comparison between the soil of our vegetable garden and that of a natural system | GA13, GB6, GB10, GC6 |
| living beings | What biodiversity of species exists in the garden? How can we improve it? | GA8, GB2, G7C |
| living beings, matter | What effect do mineral salts have on soil fertility? | GA5, GB11, GC2 |
| living beings | What biodiversity of macrofauna does the vegetable garden have? Estimation of the biodiversity of macrofauna | GA1,GB3, GC11 |

### 3.2. Research Focus

A descriptive and interpretive study, the main objective of which is to understand the phenomena that constitute the educational experience [61], was carried out. It was conducted by analysing the content of the reports the students had to prepare. A mixed qualitative and quantitative approach [62] was used. An instrument of categorising and a rubric were employed, and both were adapted before undertaking the study.

### 3.3. Data Collection Instrument

The investigation reports prepared by each working group were used to collect data. The reports were structured based on different sections established by the teachers:

1.  Introduction: the subject of the investigation, its relationship with the early childhood education curriculum, and the main objective of the investigation.
2.  Theoretical framework: in-depth information related to the topic to understand and contextualise the problem question posed.
3.  Methodology: the problem question, the hypothesis to test, and the investigation design.
4.  Results and data analysis: the findings of the investigation, and the connection with the hypotheses formulated in the study. In this section, the students were asked to include graphs, tables, or any type of representation that would help explain the results obtained when verifying the hypotheses and answering the problem question.
5.  Final conclusions and proposals for improvement: final reflection, proposals for improvements, analysis of the limitations, and the didactic potential of the investigation.
6.  Bibliography using the APA format.

*3.4. Data Analysis*

A total of 37 investigation reports of variable length (between 8 and 22 pages) of the 37 working groups were analysed: 13 from class-group A, 12 from class-group B and 12 from class-group C. All the reports were analysed to assess the enquiry competency level of the students. To this end, an instrument proposed by Ferrés et al. [29] was employed. It is an adaptation of the instrument previously designed by Tamir et al. [63] called Practical Test Assessment Inventory (PTAI), which consists of a total of 21 categories. Each category addresses and determines the students' investigative skills based on determining factors in the enquiry process. Based on this instrument, Ferrés et al. [29] propose another instrument called New Practical Test Assessment Inventory (NPTAI) to evaluate the general processes of an open and autonomous scientific enquiry, reducing the number of categories from 21 to 7: (1) identification of problems that can be investigated, (2) formulation of hypotheses, (3) identification of variables, (4) investigation planning, (5) data collection and processing, (6) data analysis and obtaining reasoned conclusions and (7) meta-reflection.

In this study, which follows a guided enquiry, some modifications to the instrument of Ferrés et al. [29] were put forward. The seven initial categories that make up the NPTAI were reduced to five. The category corresponding to identifying problems that can be investigated was eliminated, since the problems to investigate were provided by the teachers. The meta-reflection category was also deleted, as it was not an aspect to be included in the investigation report. The category corresponding to the formulation of hypotheses was modified, since this requirement in the form of deduction was not considered. It is complex and requires training that was not possible to work on with the students in question.

Once the NPTAI (adapted to a guided enquiry) was applied to the data provided in the group investigation reports, the results were recoded to establish the ECL in accordance with Ferrés et al. [29]. This process was performed by means of a rubric proposed by the authors of this paper. It enables assessing the investigation reports prepared by the groups in a quantitative manner. The rubric includes five levels of competency. Based on the assessment obtained, the students were categorised into levels that went from unscientific to the "enquirer" level, a high enquiry level. The rubric was adapted to adjust the categories to the EBS implemented in the DNE subject.

After adapting the instruments, the reports were analysed following a method of inter-rater analysis between the two researchers of the study in accordance with similar research studies [5,64]. Each group investigation report was first categorised separately. The reports were then shared to determine coincidences and resolve possible discrepancies. Finally, an analysis of frequencies and percentages was performed to quantify the enquiry competency levels established. They are represented in an Excel file.

## 4. Results

Table 2 shows the results obtained from the NPTAI instrument for each of the reports prepared by the different groups (A, B and C) corresponding to the categories and levels established.

With regard to the formulation of hypotheses, in 32.4% of the investigation reports, the hypotheses considered were unrelated to the problem to investigate (level 1). A total of 37.8% of the groups formulated hypotheses related to the problem for investigation, but none of them followed a deductive approach (level 3). Some examples of this type are: "without plant cover on the earth, a larger quantity of soil will come off" (GA2), or "air pollution makes it difficult for plants to grow" (GA3).

As far as the identification of variables is concerned, 43.2% of the groups (N = 37) indicated the variables, but without identifying the type of variables (level 3). However, they were in accordance with the hypotheses formulated. A total of 27% of the reports was quantified under level 1. This corresponds to the groups that did not identify any variables or did not know how to specify them despite having considered them in the design.

**Table 2.** Values obtained from the NPTAI instrument adapted to this study.

| | Levels | Group A | Group B | Group C | Total | Total % |
|---|---|---|---|---|---|---|
| Formulation of hypotheses | 0 | 1 | 1 | 2 | 4 | 10.8 |
| | 1 | 5 | 3 | 4 | 12 | 32.4 |
| | 2 | 3 | 1 | 3 | 7 | 18.9 |
| | 3 | 4 | 7 | 3 | 14 | 37.8 |
| | 4 | 0 | 0 | 0 | 0 | 0.0 |
| | **Levels** | **Group A** | **Group B** | **Group C** | **Total** | **Total %** |
| Identification of variables | 0 | 1 | 2 | 1 | 4 | 10.8 |
| | 1 | 4 | 3 | 3 | 10 | 27.0 |
| | 2 | 3 | 0 | 2 | 5 | 13.5 |
| | 3 | 5 | 6 | 5 | 16 | 43.2 |
| | 4 | 0 | 1 | 1 | 2 | 5.4 |
| | **Levels** | **Group A** | **Group B** | **Group C** | **Total** | **Total %** |
| Planning of the investigation | 0 | 1 | 1 | 0 | 2 | 5.4 |
| | 1 | 1 | 3 | 2 | 6 | 16.2 |
| | 2 | 4 | 2 | 4 | 10 | 27.0 |
| | 3 | 7 | 5 | 6 | 18 | 48.6 |
| | 4 | 0 | 1 | 0 | 1 | 2.7 |
| | **Levels** | **Group A** | **Group B** | **Group C** | **Total** | **Total %** |
| Data collection and processing | 0 | 1 | 0 | 0 | 1 | 2.7 |
| | 1 | 3 | 4 | 4 | 11 | 29.7 |
| | 2 | 8 | 3 | 4 | 15 | 40.5 |
| | 3 | 1 | 5 | 4 | 10 | 27.0 |
| | 4 | 0 | 0 | 0 | 0 | 0.0 |
| | **Levels** | **Group A** | **Group B** | **Group C** | **Total** | **Total %** |
| Data analysis and obtaining conclusions | 0 | 1 | 0 | 0 | 1 | 2.7 |
| | 1 | 3 | 7 | 3 | 13 | 35.1 |
| | 2 | 5 | 2 | 6 | 13 | 35.1 |
| | 3 | 4 | 2 | 2 | 8 | 21.6 |
| | 4 | 0 | 1 | 1 | 2 | 5.4 |

Appropriate variables were identified and defined in accordance with the hypotheses formulated in only 5.4% of the reports analysed. An example was provided by GB3, which put forward as a hypothesis: "the vegetable garden has a low biodiversity of macrofauna." This group defined the variables of their experiment as the number of arthropods, earthworms and molluscs situated in a 10 × 10 cm quadrant with a 10 cm depth in the different beds of the vegetable garden.

With respect to planning the investigation performed, 48.6% of the reports provided an adequate methodological design to check the hypotheses formulated. However, they did not consider controls or replicas within the design (level 3). Only 2.7% of them followed a methodological design appropriate for testing the hypotheses, proposing controls of the experimental situation. This was the case of GA3. They investigated the effect of air pollution on the growth of aubergine plants by covering leaves with a layer of moisturising cream. They used leaves with moisturiser and leaves without control: "Several leaves of the plant were covered with moisturising cream, and we left some without moisturiser to be able to compare the difference between the two."

In 40.5% of the reports, the data collected and processed contained errors and inaccuracies, or there was a lack of understanding of the experimental procedure. There was

no relationship between the data collected and the hypotheses suggested, although those groups processed the data properly, and represented them graphically (level 3). Not a single group collected data that enabled them to acquire a thorough understanding of the experimental techniques and measurements. None of the groups processed the data properly by means of graphs or by using and comparing data.

Only 5.4% of the research reports analysed included a well-founded data analysis and evidence-based conclusions, taking into account the information presented in the theoretical part of the work. A total of 35.1% of the groups presented a poor analysis, and their conclusions were not based on the data obtained (level 1), or their conclusions were a summary of the results, but without interpreting the data (level 2). An example was provided by GA8. They concluded by summarising the results obtained: " . . . the results were rather unbalanced, since there is a diversity of plants, but not of animals in the vegetable garden. It needs to be stressed that this does not mean that the results are not effective, since they are reliable." Other groups presented positive aspects of the method used for their experiment in their conclusions, but without indicating explicit conclusions based on their results: "The decomposition column is a great instrument for creating our own compost, since we use the waste we usually throw away every day. It is of great importance to know the usefulness of the 3Rs because people consider that materials such as plastic or cardboard can be recycled to be reused. However, while the same is true for organic matter, we do not think about it. It can be turned into fertiliser that allows feeding the plants and creating more food" (GC9). GA6 concluded that: "at the end of the investigative process, it was observed that applying fertiliser is not essential for proper germination and subsequent growth, since each plant has had a different development." However, an interpretation of the reason for these results considering its theoretical framework is not provided.

As a result of the previous analysis, Figure 2 shows the data of the total distribution of the groups in each of the competencies associated with the enquiry process. The ECLs show a predominance of students located at the "enquirer" level (59.5%), followed by the incipient level (21.6%) and the insecure enquirer (13.5%). Only 2.7% of the students was situated in the lowest (pre-scientific and non-scientific) levels.

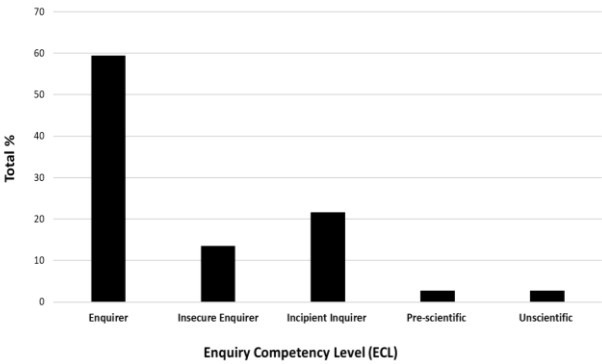

**Figure 2.** Representation of the total distribution (%) of the groups based on the enquiry competency levels in accordance with the ECL instrument.

## 5. Discussion

There is no doubt that the experience trainee teachers have during initial teacher training clearly influences their future teaching practice [65]. Initial teacher training should be aimed at training teachers capable of designing proposals that allow for developing competencies and skills established in the curriculum.

In our study, the data of the total distribution of the groups in each of the categories associated with the enquiry process show the aspects in which future teachers have greater or lesser difficulty when carrying out investigations. These results are similar to those of other studies [29,66–68]. Students at different educational stages experience significant

difficulties in each of the phases of the investigative process, mainly in the first phases. In this case, they correspond to the hypothesis formulation of the problem questions contextualised in the UEG, and the identification of variables. More than 50% (N = 37) of the reports analysed were situated in the three lowest levels of the instrument used, which shows an important weakness when formulating hypotheses. This deficiency seems to affect the rest of the scientific skills analysed. As Subagia and Wiratma [68] point out, a hypothesis is a temporary answer based on a problem question that is posed and that needs to be tested by evidence that is generally obtained from the experiment. For this reason, the rest of the scientific processes involved in an enquiry (design, data collection, analysis, drawing conclusions and communication of results) are carried out on the basis of the hypotheses formulated. Consequently, as shown by the results obtained in this study, the groups performed at lower levels in the rest of the scientific skills considered. This means there is room for improvement in these learning processes.

Careful thought should be given to how we learn the skills linked to the scientific process. These are essentially acquired in school contexts, and they require a conscious and explicit process. According to Mercer et al. [69], teachers do not usually offer this type of guidance to their students, who need help to learn and develop these skills. Likewise, formulating hypotheses requires considering aspects of specific writing that make it possible to use a deductive approach. In our case, this was only present in 37.8% of the reports.

As far as the identification of variables is concerned, the trend shown is very similar to that observed in the formulation of hypotheses. Thus, 43.2% of the groups were able to point out variables, but without specifying the type of variables, as shown by the predominance of lower levels for this scientific skill. Almost 50% of the groups did not identify variables or did not consider them in their designs. Similar results were obtained by Aydoğdu [67], who states that future teachers have difficulties in accurately identifying the dependent, independent and control variables. In our case, variables were identified and defined in accordance with the hypotheses formulated in only 5.4% of the reports, which shows a very low level in this skill.

As for the designs used, in many cases the teachers had to guide the students for the designs to be repeatable, simple and, above all, viable. Designs based on standard sampling methods stood out, such as the use of quadrants to quantify diversity and macrofauna, soil models with different kinds of plant protection to check erosion, or the use of decomposition columns. The methodological problem found in most of the groups was, as previously mentioned, a consequence of the difficulty in identifying and defining the variables involved in the experiment, taking into account the (qualitative or quantitative) nature of these variables. The students also experienced difficulties when using the control condition in order to compare the results obtained and draw clear conclusions. This is a complicated concept for students to understand [70,71]. Perhaps, in this case, it was due to the fact that the students found it difficult to understand the scientific methodology itself, which they consider to be a process restricted to the field of scientists. They believe specific materials and equipment are necessary to use it. Another weak point observed during the development of the designs was the idea that they had to be adapted to the early childhood education stage. The students were not fully aware of the fact that the investigation carried out was a personal experience in which they were active subjects in the learning process. The investigation conducted could be introduced as an enquiry strategy in their future teaching practice.

With regard to the analysis of the data obtained, the greatest difficulty was perceived at the time of recording or collecting data, which was done in an incomplete way. This may be because the students did not know which variables to record. Some groups provided data that did not consider the hypotheses formulated, which shows a lack of understanding of the procedures. Yet, many of the groups created tables and graphs and presented the results obtained in a global manner.

## 6. Conclusions

The work presented shows certain difficulties the students, future early childhood teachers, experience in the development of investigative skills such as formulating hypotheses and identifying variables. This stresses the need to carry out activities aimed at achieving a greater prominence of scientific activity in the classroom, contributing to increasing student motivation and engagement.

The skills involved in scientific processes are an inseparable component of enquiry-based science education [72]. They are especially important for citizens to achieve scientific literacy [73]. According to Abungu et al. [74], scientific skills are essential to be able to solve problems or carry out scientific experiences that are transferable to society and that imply being able to decide to express one's opinion and think critically. We hence conclude that the EBS in the context of the UEG provides an ideal scenario to consider different questions for investigation related to several topics of the curriculum: soil, fertility, cycles of organic matter, biodiversity and irrigation systems. Taking these topics as a starting point, although there were limitations, the students designed small investigations using the resources and means available in the vegetable garden.

A second conclusion is that the results show that the students need to improve their enquiry competency level. The proposal presented in initial teacher training is an opportunity for future teachers to use scientific practices. This means they need to be introduced to developing scientific skills that can be applied in school contexts. We consider that, by combining enquiry and the vegetable garden, a great obstacle is overcome. Cañal et al. [75] detected this barrier in active teachers when implementing enquiry strategies in schools. They point out the lack of personal experience (as students or future teachers) of real processes of school investigation, of theoretical-practical references and of materials and procedures characteristic of science.

Finally, as an educational implication, we consider it necessary to reflect on our own actions in the classroom. The role we play as teachers in making each step followed in the enquiry process more explicit is fundamental. It is necessary to dedicate more time and to provide more feedback to the groups of students when they formulate hypotheses and identify variables, since the first steps are key for the development of their enquiry competency level.

**Author Contributions:** Conceptualization, L.A. and B.G.-C.; methodology, L.A. and B.G.-C.; formal analysis, L.A. and B.G.-C.; investigation, L.A. and B.G.-C.; resources, L.A. and B.G.-C.; data curation, L.A. and B.G.-C.; writing—original draft preparation, L.A. and B.G.-C.; writing—review and editing, L.A. and B.G.-C.; supervision, L.A.; funding acquisition, L.A. and B.G.-C. All authors have read and agreed to the published version of the manuscript.

**Funding:** This research was funded by Universidad de Cádiz, Spain.

**Data Availability Statement:** Not applicable.

**Acknowledgments:** We would like to thank Ann Swinnen for her useful feedback and comments.

**Conflicts of Interest:** The authors declare no conflict of interest.

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
