# Peer review of "Promoting Enquiry Skills in Trainee Teachers within the Context of the University Ecological Garden"

_education, doi:10.3390/educsci12030214_

Round 1

Reviewer 1 Report

The subject of the paper is relevant. The title is appropriate to the content of the work. The summary is relevant. There is consistency between the approaches set out in the introduction. The methodology is adequately described. A good description of the results is made. The conclusions respond to the objectives of the work.

To be published, at least the following corrections must be addressed

.

Introduction

Lines 36-37 must refer to Spain.

Theoretical Framework

In Lines 163-165 do not give identity data of the authors.

Materials and methods

Improve description of the participants. How many groups? 38 appear but there should be 37. Correct table 1. The CG4 group appears in two themes. Correct group name G4C, G5C, G6C.

Data analysis

On line 240, review the data as they conflict with those provided in Table 1. It says “A total of 36 investigation reports of variable length (between 8 and 22 pages) of the 36 working groups were analysed: 11 from class-group A, 13 from class-group B, and 12 from class-group C”. However, Table 1 shows 38 groups. One repeated on GB4 to which two subjects are assigned. Table 1 shows that in group A there are 13 groups, 12 in group and 12 in group C.

From table 2 it can be deduced that 37 investigations reports are analysed. Review this topic because if not, it affects the data that is presented.

Author Response

Dear Editors,

We greatly appreciate the reviewers’ insightful comments and suggestions. We consider their contributions have helped to considerably improve the quality of our paper.

Reviewer’s comment

Introduction

Lines 36-37 must refer to Spain.

Authors’ answer

We have changed “at the national level” for “in Spain.”

Reviewer’s comment

Theoretical Framework

In Lines 163-165 do not give identity data of the authors.

Authors’ answer

We have added information on the authors to support the statement made in lines 163-165. We have checked the numbering of all the references used in the article.

Reviewer’s comment

Materials and methods

Improve description of the participants.

Authors’ answer

We have added how the different participants have been organised into groups. Should there be any additional information that we have overlooked, we would appreciate it if you could point it out to us.

Reviewer’s comment

Materials and methods

How many groups? 38 appear but there should be 37. Correct table 1. The CG4 group appears in two themes. Correct group name G4C, G5C, G6C.

Authors’ answer

We have checked this information, and there are indeed 37 groups (13 in group A, 12 in group B and 12 in group C). The data shown in Table 1 have been corrected. We have also changed the data in the abstract and throughout the text. Group GC4 group had been duplicated. We have corrected this.

Reviewer’s comment

Data analysis

On line 240, review the data as they conflict with those provided in Table 1. It says “A total of 36 investigation reports of variable length (between 8 and 22 pages) of the 36 working groups were analysed: 11 from class-group A, 13 from class-group B, and 12 from class-group C”. However, Table 1 shows 38 groups. One repeated on GB4 to which two subjects are assigned. Table 1 shows that in group A there are 13 groups, 12 in group and 12 in group C.

Authors’ answer

We have checked this information, and there are indeed a total of 37 reports. The mistake has been corrected. The mistake pointed out in the number of groups in Table 1 has also been corrected.

Reviewer’s comment

From table 2 it can be deduced that 37 investigations reports are analysed. Review this topic because if not, it affects the data that is presented.

Authors’ answer

We have verified that there are indeed 37 reports. The analysis has been carried out based on this number of reports, which corresponds to the data presented.

Reviewer 2 Report

Using as a framework structure the inquiry-based strategy (EBS) and the University Ecological Garden (UEG) as a pedagogical tool, the authors analyse (NPTAI instrument) the work carried out by prospective teachers in order to identify the skills acquired on the investigative process (phases, variables, data collection, ... ), measured by the enquiry competence level (ECL).

The paper is of interest and it is well written. In any case I suggest an extra text editing checking.

The introduction covers the importance of children developing cognitive-linguistic skills and, therefore for prospective teachers learning how to teach them, but I miss a further elaboration of teachers' knowledge or the competences needed linked to the enquiry process. As these are the subject of the research, they deserve a paragraph or two more, and also I would expand on what we (as teacher trainers) specifically need to improve in the training of prospective teachers.

(I miss a point in the short explanatory title of Table 2)

Author Response

Dear Editors,

We greatly appreciate the reviewers’ insightful comments and suggestions. We consider their contributions have helped to considerably improve the quality of our paper.

Reviewer’s comment

The paper is of interest and it is well written. In any case I suggest an extra text editing checking

Authors’ answer

Thank you for your comment. The text has been revised by a native English speaker specialised in proofreading scientific articles. We trust the modifications made have improved the readability of the paper.

Reviewer’s comment

The introduction covers the importance of children developing cognitive-linguistic skills and, therefore for prospective teachers learning how to teach them, but I miss a further elaboration of teachers' knowledge or the competences needed linked to the enquiry process. As these are the subject of the research, they deserve a paragraph or two more, and also, I would expand on what we (as teacher trainers) specifically need to improve in the training of prospective teachers.

Authors’ answer

Following your recommendation, information regarding the knowledge teachers should have when undertaking a successful enquiry process in the classroom has been included in the introduction.

Reviewer’s comment

I miss a point in the short explanatory title of Table 2

Authors’ answer

We are afraid we fail to understand what you mean. Could you please be more specific?